# Ultra-NeRF: Neural Radiance Fields for Ultrasound Imaging

**Magdalena Wysocki**[*1]                                          MAGDALENA.WYSOCKI@TUM.DE
**Mohammad Farid Azampour**[*1,2]                                  MF.AZAMPOUR@TUM.DE
**Christine Eilers**[1]                                            CHRISTINE.EILERS@TUM.DE
**Benjamin Busam**[1,3]                                            B.BUSAM@TUM.DE
**Mehrdad Salehi**[1]                                              MEHRDAD.SALEHI@TUM.DE
**Nassir Navab**[1]                                                NASSIR.NAVAB@TUM.DE

[1] *Computer Aided Medical Procedures & Augmented Reality, Technische Universität München*

[2] *Department of Electrical Engineering, Sharif University of Technology*     [3] *3Dwe.ai*

**Editors:** Accepted for publication at MIDL 2023

## Abstract

We present a physics-enhanced implicit neural representation (INR) for ultrasound (US) imaging that learns tissue properties from overlapping US sweeps. Our proposed method leverages a ray-tracing-based neural rendering for novel view US synthesis. Recent publications demonstrated that INR models could encode a representation of a three-dimensional scene from a set of two-dimensional US frames. However, these models fail to consider the view-dependent changes in appearance and geometry intrinsic to US imaging. In our work, we discuss direction-dependent changes in the scene and show that a physics-inspired rendering improves the fidelity of US image synthesis. In particular, we demonstrate experimentally that our proposed method generates geometrically accurate B-mode images for regions with ambiguous representation owing to view-dependent differences of the US images. We conduct our experiments using simulated B-mode US sweeps of the liver and acquired US sweeps of a spine phantom tracked with a robotic arm. The experiments corroborate that our method generates US frames that enable consistent volume compounding from previously unseen views. To the best of our knowledge, the presented work is the first to address view-dependent US image synthesis using INR.

**Keywords:** ultrasound, neural radiance fields, implicit neural representation

## 1. Introduction

3D visualization of an anatomy significantly improves our understanding of the underlying pathology, however, most US machines used in practice deliver only a single cross-sectional view of an anatomy at a time. Sonographers, through extensive training and clinical expertise, fuse these 2D scans into a 3D model in their minds. The anisotropic nature of US imaging contributes to the increased difficulty of this task. Since an image of a specific region in the patient's body depends on the probe position, a mental 3D model is constantly updated with images that may carry contradicting information for the same region. Nevertheless, a trained operator approaches this problem effortlessly owing to the consciousness of the anatomy and the effect of the probe position on its 2D representation. However, this manual visual analysis remains expensive and error-prone (Lyshchik et al., 2004; Kojcev

---

[*] Contributed equally

et al., 2017; Krönke et al., 2022). A system that can represent the underlying 3D geometry of an anatomy based on multiple US scans can reduce the error rate.

Much research has recently been devoted to utilizing 3D US as an explicit way of modeling the geometry. 3D US is used in diagnostic applications, as well as interventional radiology. 3D US volumes are conventionally generated using special wobbler probes, 2D transducers, or tracked probes to compound a 3D volume from 2D slices (Busam et al., 2018). In the last decade, new approaches such as computational sonography (Hennersperger et al., 2015), sensorless 3D US (Prevost et al., 2017), and deep learning-based image formation techniques (Simson et al., 2018) have aimed at improving the 3D compounding quality of this portable and affordable modality. However, none of these approaches can properly model the underlying scene. When looked at the compounded volume from different views, the resulting B-mode image is not a plausible one. As an example, when slicing from a novel view, acoustic shadows might be displayed where there should be none or an anatomy is shown where no rays could possibly reach it from this novel view. However, with a correct modeling, one can generate plausible B-mode scans from any arbitrary view. Our proposed approach focuses on learning the underlying geometry using 2D US images from different viewpoints. Our method enables us to generate physically plausible B-mode scans from novel views and introduces a new implicit US representation for the medical image processing community to explore.

Although viewing-direction dependency is a prominent characteristic of US imaging, it is not a unique property of US. To some extent, a similar phenomenon characterizes natural images. For instance, since the non-Lambertian assumption does not hold for most real world objects, appearance due to reflections might be inconsistent between views (Gao et al., 2022). 3D scene reconstruction from a set of 2D view-dependent observations has been hence extensively studied (Seitz and Dyer, 1999; Niemeyer et al., 2020). An important aspect of any reconstruction method is a scene representation, which can be either explicit (e.g. volumetric grids), or implicit (e.g. implicit functions). Implicit scene representations, such as (truncated) signed distance functions ((T)SDFs) (Newcombe et al., 2011) represent a 3D scene as a function. Since neural networks are universal function approximators, they can be used to parametrize an implicit representation (Tewari et al., 2022). This fact has been a basis of a recent development in neural volumetric representation. In particular, Neural Radiance Fields (NeRF) emerged as a new, potent method for generating photorealistic, view-dependent images of a static 3D scene from a collection of pose-annotated images (Mildenhall et al., 2021). In computer vision, NeRF became a baseline for for various research directions such as dynamic scenes (Park et al., 2021), large scale scenes (Rematas et al., 2022), or scene generalization (Yu et al., 2021). Moreover, as presented in iNeRF (Yen-Chen et al., 2021) representing a 3D model as a neural network provides a reference for 6DoF pose estimation, which potentially can find an application in US tracking. The idea behind NeRF, however, was primarily developed for natural image synthesis and takes advantage of established methods from computer graphics. In this paper, for the first time, we propose an implicit neural representation with physics-based rendering for US imaging exemplified with NeRF that facilitates the synthesis of B-mode images from novel viewpoints. Our contributions are as follows:

- a method that synthesises accurate B-mode images by learning the view-dependent appearance and geometry of a scene from multiple US sweeps;

- a physically sound rendering formulation based on a ray-tracing model, which considers the isotropic tissue characteristics important to US;
- open source datasets[1] comprising multiple tracked 2D US sweeps with highly accurate pose annotations and different viewpoints.

In our experiments, we use synthetic liver and spine phantom data. We evaluate our method quantitatively and qualitatively. In particular, we demonstrate the importance of rendering based on the physics behind US imaging and the shortcomings of learning an INR without considering view-dependent changes to the observed scene. To the best of our knowledge, this paper presents a new implicit neural representation for US that for the first time considers the anisotropic characteristics of US.

## 2. Related Work

Implicit representations in the form of (T)SDFs have been used for implicit geometric reconstruction (Newcombe et al., 2011). Recently, INR has been proposed to express signals as a neural network (Sitzmann et al., 2020) which can be seen as a universal function approximator, which represents a scene as a continuous function parameterised by its weights. As a consequence, it allows for a mapping from a 3D continuous coordinate space to intensity to store information about a 3D scene. As presented by Gu et al. (2022), we can exploit INR models to represent a 3D US volume learnt from a set of 2D US images. However, parametrizing an US volume using a 3D continuous coordinate space does not address a viewing direction impact on the observation. The progress in neural continuous shape representation sparked interest in their application to photorealistic novel view synthesis. In particular, in a seminal work introducing NeRF (Mildenhall et al., 2021), the authors propose a framework that combines neural representation of a scene and fully differentiable volumetric rendering. In NeRF, the representation of a scene is expressed by a fully-connected neural network. The network maps a 5D vector (a spatial location $\mathbf{x}$ and viewing direction $\mathbf{d}$) to volume density $\sigma$, and radiance $\mathbf{c}$. To learn this mapping a per-pixel camera ray is defined as $\mathbf{r}(t) = \mathbf{o} + t\mathbf{d}$ with the camera origin $\mathbf{o}$ in the center of the pixel defining the near plane. The final colour value of each pixel is defined by following formulas:

$$C(\mathbf{r}) = \int_{t_n}^{t_f} T(t)\sigma(t)c(t)dt \qquad (1)$$

$$\text{where } T(t) = \exp - \int_{t_n}^{t_f} \sigma(\mathbf{r}(s))ds \qquad (2)$$

The volume rendering integral in Equation (1) accumulates ray-traced radiance field from near $t_n$ to far plane $t_f$, with each position contributing to the final pixel color. The input of each sample is controlled by the transmittance factor $T(t)$ (Equation (2)). Finally, the rendered pixel value is compared with a value in an image using a photometric loss.

Since its introduction, NeRF-based methods have demonstrated impressive results in various fields including medical imaging. For instance, MedNeRF propose a NeRF framework to reconstruct CT-projections from X-ray (Corona-Figueroa et al., 2022), and EndoNeRF adopts NeRF for surgical scene 3D reconstruction (Wang et al., 2022).

---

1. https://github.com/magdalena-wysocki/ultra-nerf

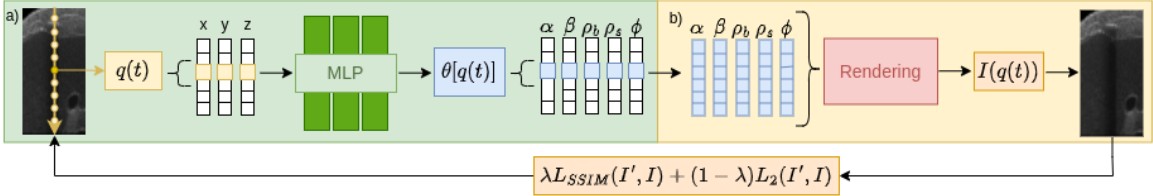

Figure 1: a) For a query point $q \in \mathbb{R}^3$ sampled along a ray, the MLP estimates a parameter vector $\theta \in \mathbb{R}^5$ from an implicit volume representation, b) from parameters at queried and preceding positions along the ray the rendering computes a per-query intensity. Resulting intensities compose an US image. The output and target frame are compared using a weighted sum of Structural Similarity Index Measure (SSIM) and Squared Error Loss (L2).

Yet, surprisingly little investigation has been done to explore the potential of neural volumetric implicit representations for medical US. One of a few studies (Yeung et al., 2021; Gu et al., 2022; Song et al., 2022) focuses on reconstruction of a spine using the NeRF algorithm (Li et al., 2021). In this paper, the authors demonstrate that NeRF can render high-quality US images. However, they apply NeRF without considering a volumetric rendering method, which respects US physics. To address this shortcoming, we reformulate the rendering step to include the underlying US physics and incorporate it into the NeRF framework.

## 3. Method

### 3.1. Background: US Physics

US images are generated by mapping reflected sounds from the tissue within a thin transversal slice of the body. As an anisotropic modality, US exhibits varying behavior in different types of tissue, resulting in direction-dependent differences in the resulting echoes. This characteristic highlights the importance of accounting for tissue anisotropy in the methods generating B-mode images. Intrinsic acoustic parameters such as travelling speed of sound, acoustic impedance, attenuation coefficient, and spatial distribution of sound scattering micro-structures are the main contributing factors affecting the sound reflection within the tissue. By knowing the mapping of these parameters in space and the wave propagation direction, one can simulate renderings of 3D US in arbitrary views (Salehi et al., 2015).

### 3.2. Ultrasound NeRF

Figure 1 presents our framework in the single-frame case. The method follows the original NeRF w.r.t its two components: a neural network (Figure 1a) and volumetric rendering (Figure 1b). The network represents a volume as a 3D vector-valued continuous function that maps a position $q = (x, y, z)$ in a Cartesian coordinate space into a parameter vector $\theta \in \mathbb{R}^5$ which elements correspond to attenuation $\alpha$, reflectance $\beta$, border probability $\rho_b$,

scattering density $\rho_s$, and scattering intensity $\phi$ and composes a final pixel intensity as outlined in Section 3.3. The parameter vector consists of isotropic physical tissue properties hence we do not provide explicit viewing directions to the network. This ensures that the regressed physical properties remain consistent between views, whereas the view-dependent changes are enforced by the rendering. Figure 2 illustrates the definition of a ray and query points. We encourage the reader to refer to Appendix A for the network details.

### 3.3. Ultrasound Volume Rendering

Our US volume rendering model builds upon a formulation presented by Salehi et al. (2015) that proposes a ray-tracing-based simulation model. The advantage of this model is its flexibility in representing US artifacts coming from backscattering effects.

For each scan-line $r$, Equation 3 defines a recorded US echo $E(r,t)$, measured at distance $t$ from the transducer, as a sum of reflected $R(r,t)$ and backscattered $B(r,t)$ energy:

$$E(r,t) = R(r,t) + B(r,t) \tag{3}$$

The reflected energy is defined by:

$$R(r,t) = |I(r,t) \cdot \beta(r,t)| \cdot PSF(r) \otimes G(r',t') \tag{4}$$

Where $\otimes$ is the convolution operator, $I(r,t)$ is the remaining energy at the distance $t$, $\beta(r,t)$ represents the reflection coefficient, and $PSF(r)$ is a predefined 2D point-spread function. G(r,t) admits 1 for points at the boundary and 0 otherwise. We compute it by sampling from a Bernoulli distribution parameterized by the border probability $\rho_b$. A probabilistic approach to the border definition reflects the network's uncertainty about interaction of the ray with a tissue border. The energy loss is traced along each scan-line, and the remaining energy $I(r,t)$ is modelled using the loss of the energy due to reflection $\beta$ at the boundaries and attenuation compensated by applying an unknown time-gain compensation (TGC) function. The final formulation for $I(r,t)$ assumes an initial unit intensity $I_0(r,0)$ and loss of energy at each step $dt$. We can further simplify the resulting equation by modeling the compensated attenuation $\alpha$ by a single parameter since TGC is a scaling factor:

$$I(r,t) = I_0 \cdot \prod_{n=0}^{t-1} [(1 - \beta(r,n)) \cdot G(r,n)] \cdot \exp^{(-\int_{n=0}^{t-1} (\alpha \cdot f \cdot dt))} \tag{5}$$

Consequently, $\alpha's$ correspond to the physical attenuation only up to an unknown scaling factor. The backscattered energy $B(r,t)$ from the scattering medium is a function of remaining energy $I(r,t)$ and a 2D map of scattering points $T(r,t)$:

$$B(r,t) = I(r,t) \cdot PSF(r) \otimes T(r',t') \tag{6}$$

The map $T(r,t)$ is learnt using a generative model inspired by (Zhang et al., 2020):

$$T(r,t) = H(r,t) \cdot \phi(r,t) \tag{7}$$

In this model, $H(r,t)$ admits 1 for a query point being a scattering point and 0 otherwise. This function is sampled from the Bernoulli distribution parameterized by scattering density

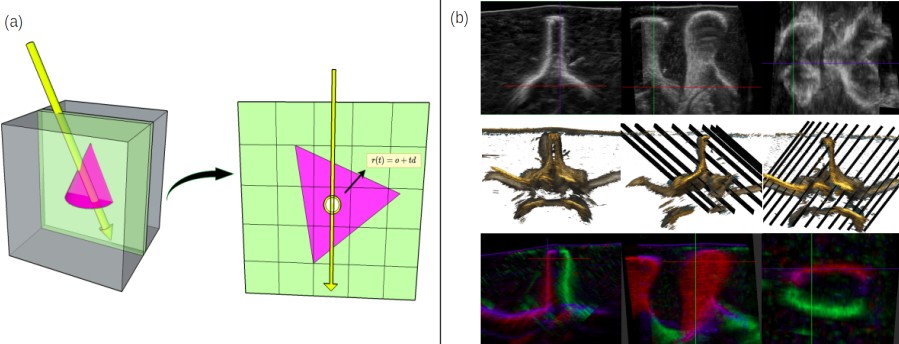

Figure 2: a) Each ray $r$ corresponds to a single scan-line with origin $\mathbf{o}$ at top of the image plane and direction $\mathbf{d}$ pointing along the scan-line. Query points are defined by their spatial location $\mathbf{r}(t) = \mathbf{o} + t\mathbf{d}$. b) Visualization of the phantom dataset: white intensities (top) from max-value compounding of all-angle images; red/blue/green intensities (bottom) compose white intensities by view angle; middle row visualizes observation directions with respect to anatomy.

$\rho_s$. It represents the uncertainty of whether the scattering effect of a scattering point is observed. The intensity of a scattering point is controlled by its amplitude $\phi$ which models sampling from a normal distribution with mean $\phi$ and unit variance. We address differentiability of the model in Appendix C.

## 4. Experiments & Results

**Data.** We acquired two types of data: synthetic and phantom B-mode images. For both datasets sweeps were recorded with different, constant perpendicular and tilt angles w.r.t acquisition direction (Figure 2b). We tested our method on 6 sweeps covering views not present in the training set. We encourage the reader to refer to Appendix B for details about the datasets.

**Quantitative Results.** Table 1 presents the evaluation of the quality of novel view synthesis as measured in terms of SSIM between synthetic and reference testing data. To analyze the effect of rendering, we compared Ultra-NeRF to an implicit neural representation model without rendering. With rendering, we achieve better or similar results on our phantom data ($\text{SSIM}_{median} = 0.54$ for tilted and $\text{SSIM}_{median} = 0.58$ for perpendicular views), whereas the method without rendering attains higher SSIM values on our synthetic dataset ($\text{SSIM}_{median} = 0.50$ for tilted, $\text{SSIM}_{median} = 0.54$ for perpendicular).

**Qualitative Results.** To showcase the advantages of our method, we evaluated against explicit 3D representation using volumetric compounding, proposed by (Wachinger et al., 2007), and a variant of our method without rendering. The results can be seen in Figure 3. We evaluated the quality of novel views by comparing slices from volumetric compounding and generated B-mode scans using Ultra-NeRF with and without the rendering function. For comparison against the explicit method, similar to Ultra-NeRF, we use multiple sweeps.

Table 1: SSIM between synthetic and reference B-mode images.

| | | with rendering | | | | w/o rendering | | | |
|---|---|---|---|---|---|---|---|---|---|
| dataset | type | median | mean | min | max | median | mean | min | max |
| liver synthetic | tilted | 0.47 | 0.45 | 0.41 | 0.60 | **0.50** | 0.51 | 0.46 | 0.59 |
| | perpendicular | 0.49 | 0.49 | 0.44 | 0.57 | **0.54** | 0.54 | 0.47 | 0.62 |
| spine phantom | tilted | **0.54** | 0.51 | 0.36 | 0.60 | 0.50 | 0.48 | 0.36 | 0.59 |
| | perpendicular | **0.58** | 0.54 | 0.42 | 0.65 | **0.58** | 0.54 | 0.41 | 0.64 |

We keep threee sweeps for testing and use the rest of the sweeps for compounding a volume. After compounding, we slice the volume based on the probe pose and orientation recorded in the test sweeps. We compare these results with the ones from Ultra-NeRF to show how explicit methods fail to preserve ultrasound's physics-based properties when generating images from novel views. Figure 3b highlights this phenomena on our spine phantom. In Appendix D, we show more results from multiple views that emphasizes the importance of physics-based rendering for learning the implicit representation and generating images from novel views.

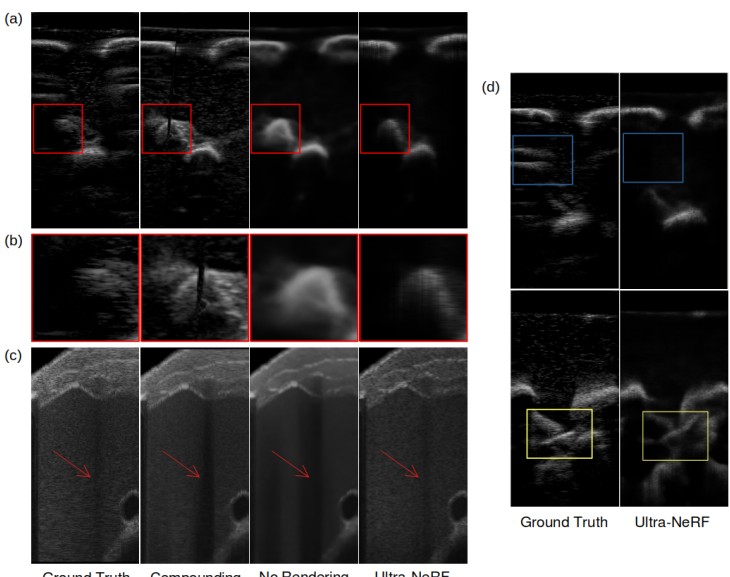

Figure 3: Evaluating performance through test-set views for our phantom (a) and synthetic (c) dataset. In (b) region marked in red on the phantom example shows an effect of occlusion that methods without rendering fail to recreate. The red arrows in (c) highlight a shadow that is reconstructed with errors when rendering is not employed. Reverberations (blue) and complex structures (yellow) are both challenging for our ray-based model (d).

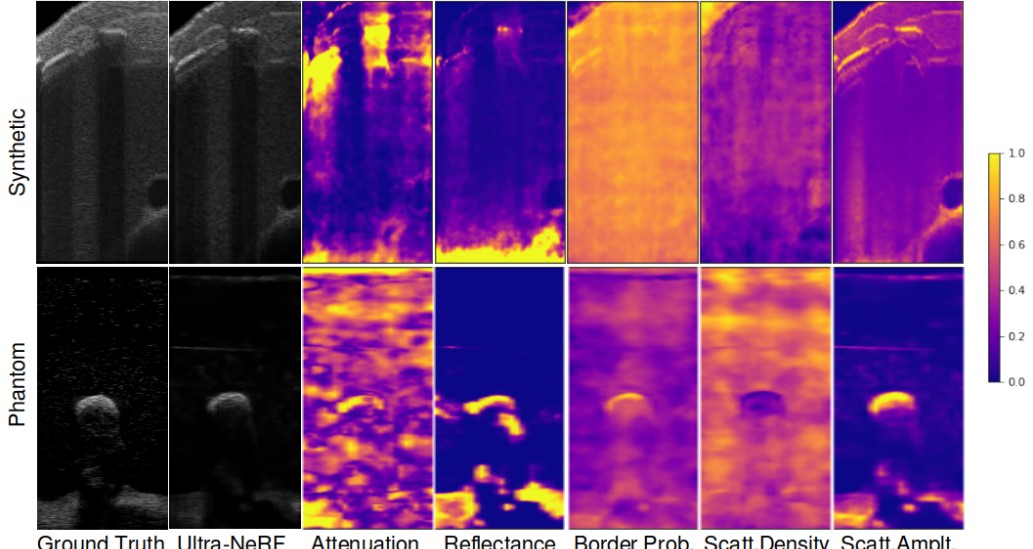

Figure 4: Example of rendering parameter decomposition for synthetic (top) and phantom (bottom) test-set data. Intermediate maps illustrate each rendering parameter. In the top row, bones exhibit higher levels of both attenuation and scattering compared to other tissue types (fat, liver, and soft tissue). In the bottom row, the bone only shows high levels of scattering. In both rows, highly reflective tissue interfaces are associated with higher reflectance. Inconsistencies in the bottom regions have no effect due to the preceding energy absorption.

## 5. Discussion & Conclusion

In this paper, we present Ultra-NeRF, a volumetric INR for US imaging from a set of 2D US B-mode scans. Unlike prior methods, our approach considers the anisotropic characteristics of US and addresses US volumetric rendering inline with the physics of US. The experiments corroborate that Ultra-NeRF incorporates information about the viewing direction into a volumetric INR, which allows for the view-dependent synthesis of US frames, resulting in high-quality B-mode images. Decomposition of a rendered B-mode in the parameter space shown in Figure 4 further illustrates that Ultra-NeRF identifies tissue characteristics leading to differences in observed intensities. For example, it correctly determines a strongly reflective structure (a rib) by regressing a region with a higher reflectance and therefore produces acoustic shadows. We encourage the reader to refer to Appendix E for additional decomposition of final renderings into rendering parameters and qualitative comparison of regressed parameters for the synthetic dataset. We propose a physically sound rendering method, however, further progress towards more realistic B-mode rendering requires addressing ray interactions and the Fresnel Effect. As shown in Figure 3, although the method learns accurate geometry, it does not allow to render complex US artifacts, such as reverberations. Additionally, to improve rendering results, future work may involve using deep learning

techniques to establish a point spread function that reflects the underlying backscattering pattern. Another potential area for future research is regularization; the decomposition into parameter space is under-constrained, thus the outcome highly depends on the initial network configuration. To the best of our knowledge, this is the first work that explores the potential of implicit neural representations for medical US by addressing a rendering method specially designed for US. Therefore, it supports progress towards integrating the implicit 3D US representation exemplified with NeRF into medical applications. We believe that this work will inspire further exploration of implicit representations in US imaging for medical purpose.

## Acknowledgments

We would like to thank Andrea Teatini and Christian Engel for providing us the phantom and their assistance with the collection of our data. The authors were partially supported by the grant NPRP-11S-1219-170106 from the Qatar National Research Fund (a member of the Qatar Foundation). The findings herein are however solely the responsibility of the authors.

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

## Appendix A. Implementation Details

### A.1. Network Architecture

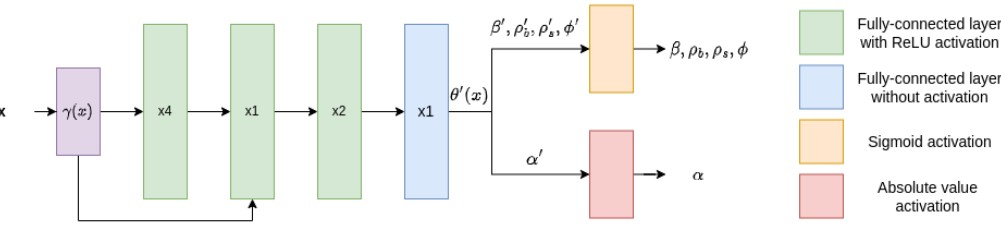

Figure 5: Schematic visualization of our MLP network.

In Figure 5 we visualize the MLP network architecture used in our experiments. In Ultra-NeRF the MLP network approximates the complex and nonlinear function that maps 3D positions to parameters values, which are used to render realistic B-mode images. By stacking multiple layers of nonlinear functions (using ReLU activations), the MLP can capture more complex and intricate features of the mapping from the Cartesian coordinates to the parameter space. On the one hand, with more layers we can approximate a more complex function. Increasing the number of layers in a model can improve its ability to approximate complex functions, but deeper models can become computationally expensive and prone to overfitting. On the other hand, shallower models may not be expressive enough to capture the complexity of the radiance field. As proposed in DeepSDF (Park et al., 2019), a balanced approach is to use an 8-layer architecture, which provides a good trade-off between model capacity and computational efficiency. Following the original network architecture we include a skip connection that concatenates the input to the fifth layer's activation. In contrast to the original NeRF architecture we do not include additional layers to include viewing direction. The viewing direction dependency of the final B-modes are therefore supported by the rendering module. To restrict the range of the final output parameters to be between 0 and 1, a sigmoid activation function is used, except for the attenuation coefficient, which can take any positive value. To handle this, an absolute value activation function is applied to the attenuation coefficient.

### A.2. Positional Encoding

The positional encoding is a function $\gamma$ that maps from a lower dimensional space to a higher dimensional space. A recent study by Rahaman and colleagues (Rahaman et al., 2019) found that deep neural networks tend to be biased towards learning lower frequency functions. However, the researchers also demonstrated that preprocessing the input data by mapping it to a higher-dimensional space using high frequency functions before passing it to the network can improve the model's ability to fit data that contains high frequency variation. This suggests that careful selection of the preprocessing techniques can mitigate the limitations of deep networks and enhance their performance in dealing with high frequency variations. In NeRF it has been demonstrated that a positional encoding improves quality of the rendered

images. Following Mildenhall et al. (2021), the function used for encoding is defined by:

$$\gamma(p) = (\sin(2^0\pi p), \cos(2^0\pi p), \ldots, \sin(2^{L-1}\pi p), \cos(2^{L-1}\pi p)) \tag{8}$$

Our experimental results provide evidence to support the claim that such an embedding is necessary for the method to learn the underlying structure. Specifically, we found that in the absence of a positional encoding, the method was not able to effectively capture the features of the ultrasound data. To facilitate our experiments, we used a default embedding size of L=10.

### A.3. Loss function

While the original NeRF approach relies solely on a photometric loss, we are enhancing it by incorporating SSIM as a similarity measure to compare the generated B-mode images with their corresponding target images. By assessing the structural and textural similarities between the two images, we can evaluate the quality and accuracy of the generated ultrasound images. In our experiments we found that using supporting SSIM by L2 loss stabilizes training and improves quality of the images. Without the L2 loss the method degenerates to predict dark patches. Our total loss value is defined as:

$$\mathcal{L} = \lambda \, \text{SSIM}(I', I) + (1 - \lambda) \sum_{i \in I} (I'(i) - I(i))^2 \tag{9}$$

The coefficient $\lambda = 0.9$ has been chosen experimentally. Although other values of the coefficient $\lambda$ may yield comparable results, we observed that lower values place the greater importance on capturing precise intensities rather than accurate structures, while higher values tend to exhibit similar limitations as a pure SSIM loss.

## Appendix B. Dataset details

Table 2: Dataset test-train split.

|  | synthetic | | phantom | |
|---|---|---|---|---|
| dataset type | training | testing | training | testing |
| number of sweeps | 4 | 3 | 8 | 3 |
| number of frames | 800 | 600 | 1200 | 300 |

### B.1. Synthetic data

We simulate B-mode images of a liver from CT images using ImFusion[3]. Each sweep comprises 2D ultrasound images and respective tracking information. Our synthetic dataset consists of seven sweeps: six with an acquisition angle tilted and one with an acquisition angle perpendicular w.r.t the surface (Figure 2b). Each sweep consists of 200 2D US images

---

3. ImFusion GmbH, Munich, Germany, software version 2.42

with respective tracking information. The tilted sweeps differ in the slope's degree and direction. Therefore, an organ is observed from different viewing angles and directions. Our frames contain occlusions caused by scanning between ribs in different directions respective to the probe direction. We use four tilted sweeps for training, totalling 800 frames, and we test on three sweeps: one perpendicular and two tilted, totalling 600 images.

### B.2. Phantom data

We acquire phantom data of a lumbar spine, gelatine-based phantom. We use a robotic manipulator (KUKA LBR iiwa 7 R800) and linear probe (Cephasonics Cicada LX ultrasound machine and Piezo Composite Linear probe) to obtain ultrasound sweeps. The position of the probe is tracked using robotic tracking. We access real-time images and tracking information using ImFusion[3]. We scan our phantom with a probe in paramedian sagittal orientation. The collected data comprise 13 sweeps with 150 frames each: six pairs of tilted sweeps and one perpendicular sweep. The trajectory of each pair of tilted sweeps is defined such that the data acquired for training completely covers tissue visible in the test data. To keep the constant spacing of images in each sweep and cover the whole testing region through training data, we reduce the number of testing frames per sweep to 100 frames. The scans occupy an area of two lumbar vertebrae. Analogously to synthetic data, the slope and direction of the probe differ between sweep pairs. As a consequence, the spinous process occludes different regions depending on the viewing direction (Figure 2). We use four tilted sweep pairs for training, totaling 1200 frames, and three sweeps for testing, totaling 300 images.

## Appendix C. Differentiability of Ultrasound Volume Rendering

The rendering module in our model is not fully differentiable because of the sampling from a Bernoulli distribution. Despite this, the loss can still back-propagate, even though the sampled variables do not contribute to the update of the network weights. To make the entire model fully differentiable, one can relax the Bernoulli distribution with a smooth approximation. Relaxed Bernoulli (Maddison et al., 2016) with low temperature value could replace the original discrete distribution to make the whole rendering differentiable. However, our initial experiments with relaxed Bernoulli show that the generated B-mode image does not change.

## Appendix D. Additional Results

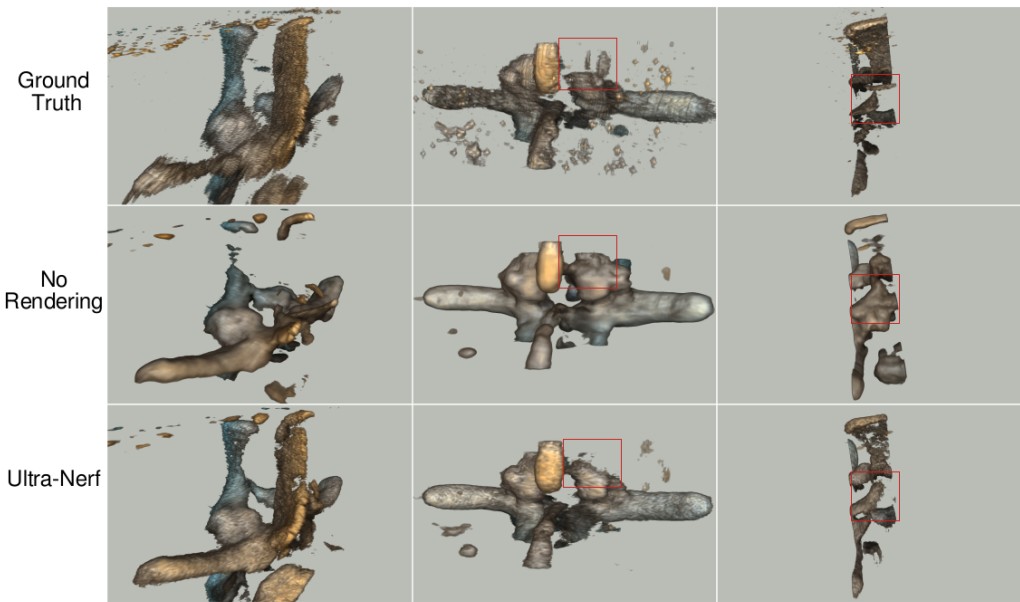

Figure 6: Compounded volumes: without rendering (middle row) the model is not aware of the viewing direction hence occluded parts of the lamina are reconstructed (red). By adding the rendering function, we introduce view-direction dependency needed to reconstruct anisotropic phenomena.

In Figure 7 and Figure 8, we demonstrate additional qualitative evaluations of our method. We compare B-modes rendered with Ultra-NeRF with ground truth B-modes, frames created with the baseline method explained in Section 4, and frames generated with INR without rendering. Next, we evaluate the consistent accuracy of rendered B-modes across the complete test-set sweep by compounding a volume using rendered frames. We compounded volumes using the compounding algorithm of ImFusion [3]. Figure 6 shows the results of the compounding. The comparison between the INR without the rendering function and Ultra-NeRF highlights the importance of rendering in representing view-dependent physical phenomena.

---

3. ImFusion GmbH, Munich, Germany, software version 2.42

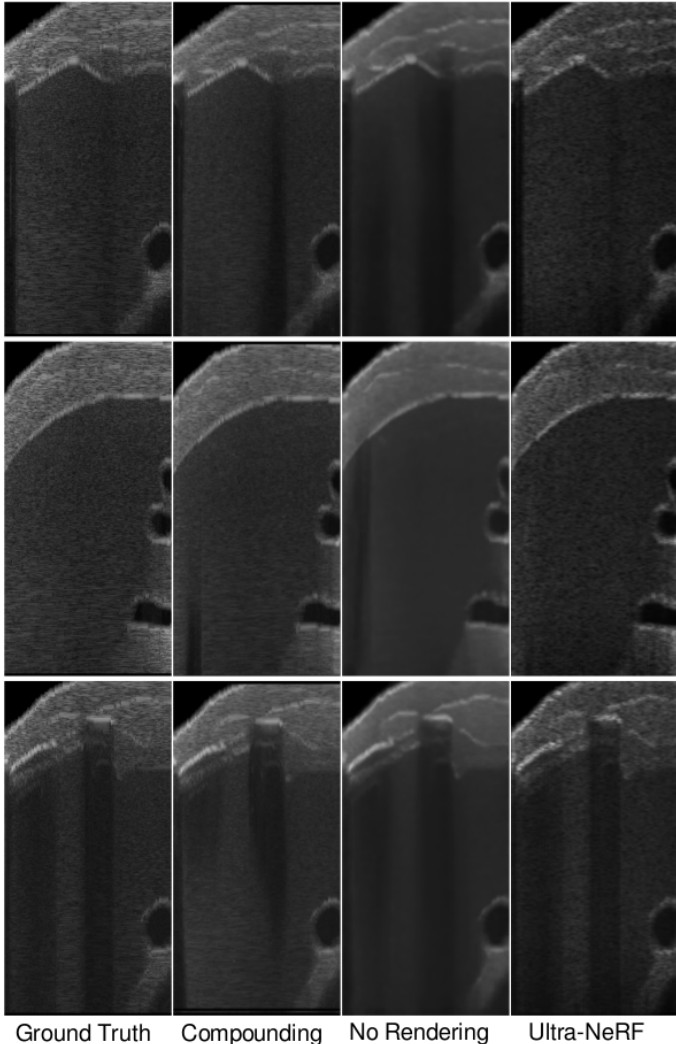

Figure 7: Comparison on 3 representative views from the synthetic test-set. We observe that Ultra-NeRF, owing to its physics-based rendering, consistently replicates view-dependent acoustic shadows that arise from highly reflective structures. In contrast, other methods that do not account for physics-based rendering cannot reproduce this phenomenon.

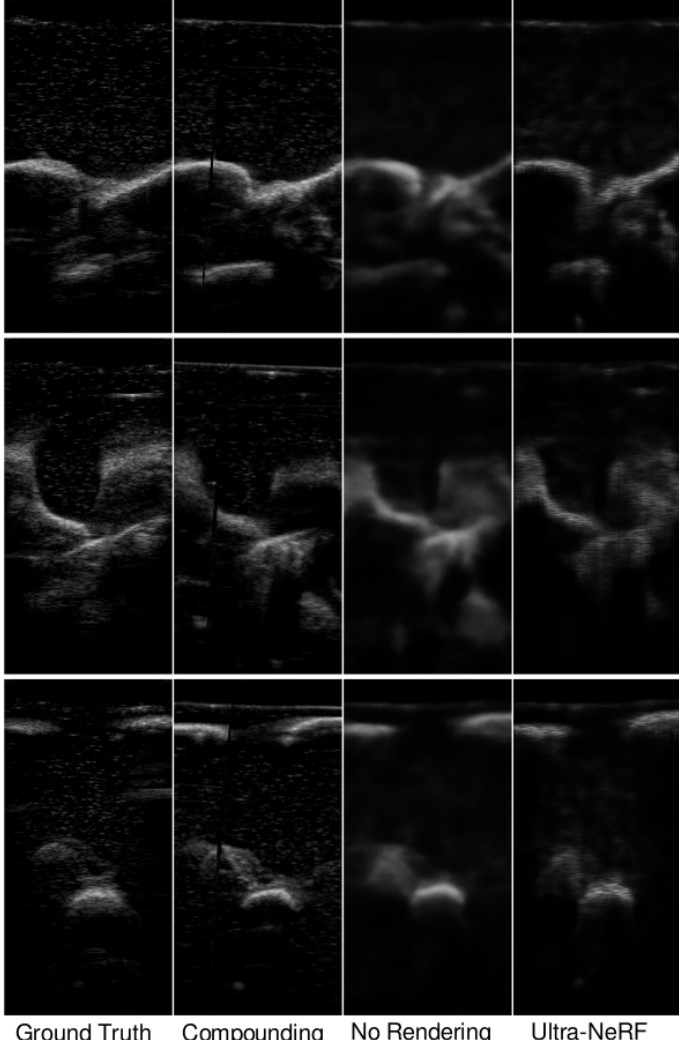

Ground Truth   Compounding   No Rendering   Ultra-NeRF

Figure 8: Comparison on 3 representative views from the phantom test-set. Our observation reveals that Ultra-NeRF, with its physics-based rendering, consistently recreates view-dependent occlusions. In contrast, other methods that do not consider physics-based rendering cannot replicate this phenomenon and produce intensities that are not visible in B-mode images.

## Appendix E. Additional Parameter Maps

The rendering model presented in Section 3.3 allows us to render B-mode images if the physical properties of tissue and the direction of the ray propagation is known. Therefore regressing physically accurate parameters is an important aspect of our method. In Figure 10 and Figure 11 we demonstrate additional examples of rendered B-modes decomposition into the rendering parameters. In Figure 9 we show a qualitative analysis of the correlation between regressed parameters and corresponding ground truth data for the synthetic dataset. Our qualitative analysis demonstrates a high correlation between the attenuation regressed by our method and the corresponding real attenuation values. Similarly, we observe that our method identifies reflective tissue interfaces present in the ground truth data. However, we found that while the scattering amplitude varies among different tissue types, the scattering density remains constant and deviates from the expected values. These differences between the regressed and expected parameter values could be attributed to the high degrees of freedom in our model. Further investigation of this aspect would be a promising avenue for future research.

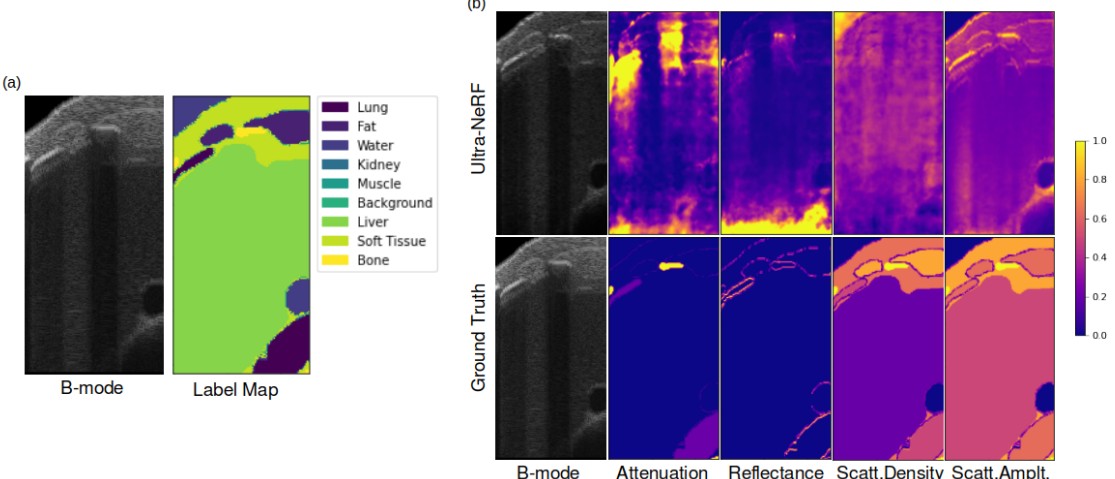

Figure 9: (a) We present tissue types visible on the B-mode. (b) Decomposition of a rendered B-mode into rendering parameters with corresponding ground truth maps.

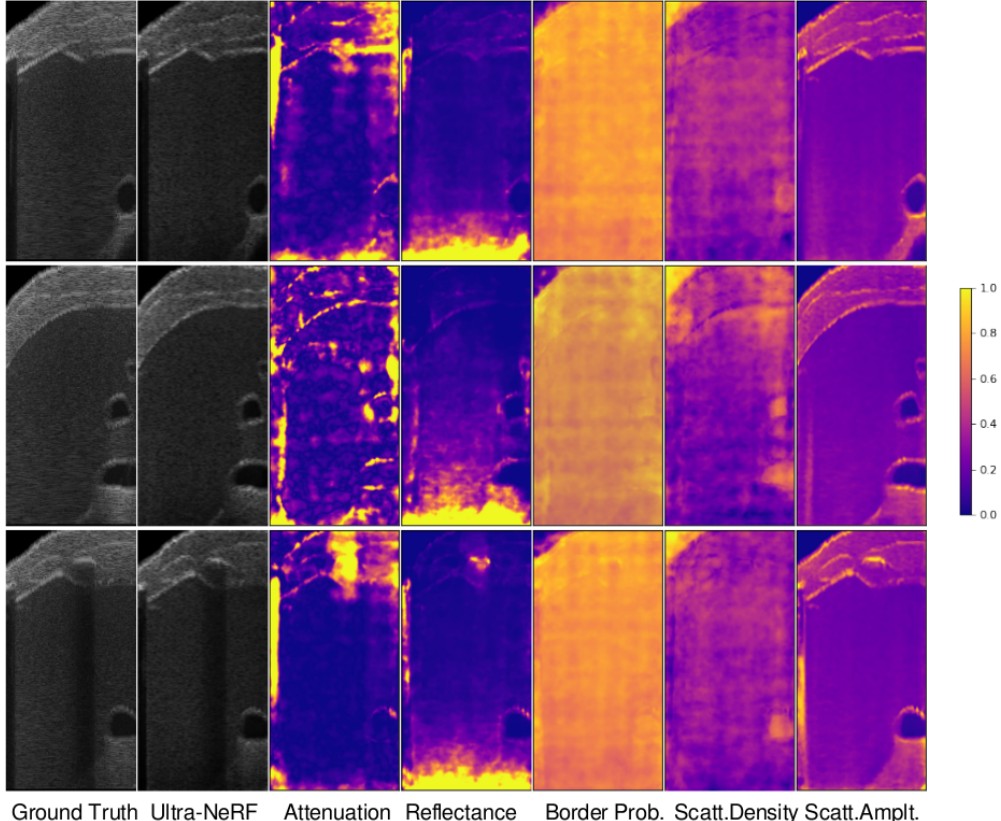

Figure 10: Examples of a B-mode image decomposition into rendering parameters for 3 representative frames from the synthetic test-set demonstrate that Ultra-NeRF consistently detects highly attenuating regions and reflective tissue interfaces, as well as differences in scattering amplitude. However, the method falls short in precisely identifying tissue interface locations and accurately retrieving the scattering density.

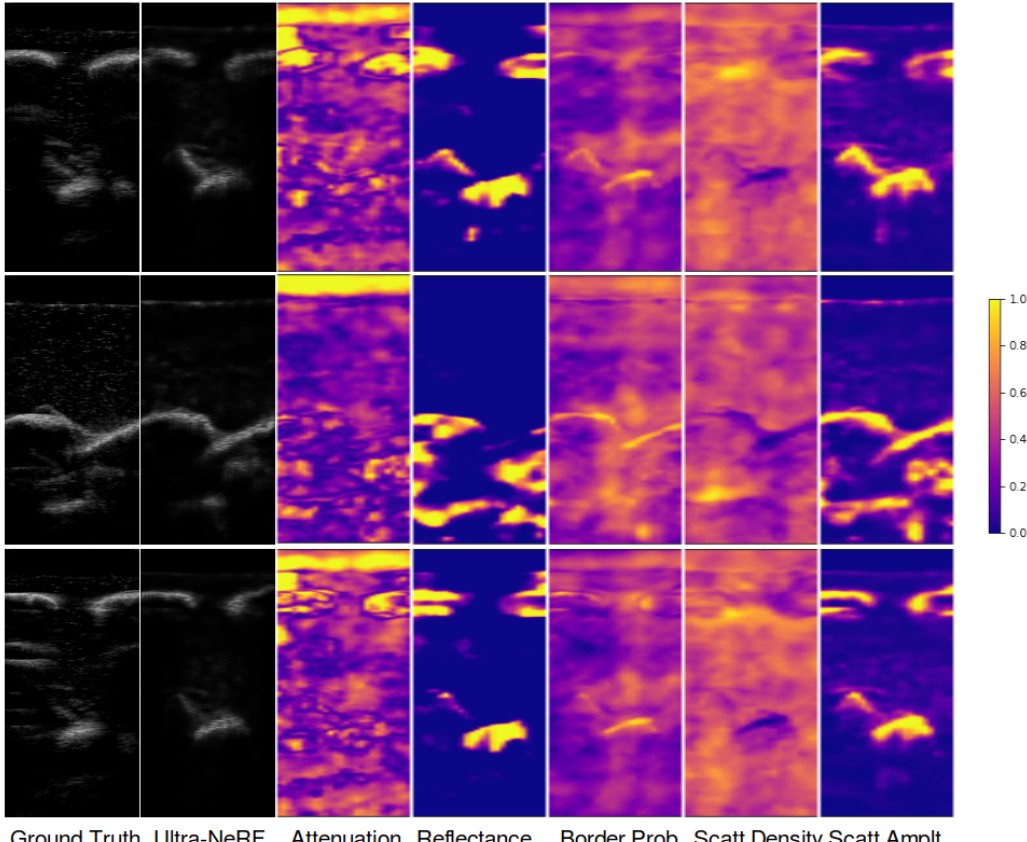

Ground Truth  Ultra-NeRF  Attenuation  Reflectance  Border Prob.  Scatt.Density  Scatt.Amplt.

Figure 11: Examples of a B-mode image decomposition into rendering parameters for 3 representative frames from the synthetic test-set illustrate that Ultra-NeRF reliably detects highly reflective regions and accurately identifies differences in the scattering amplitude. Because the two correctly regressed maps can account for the observed B-modes, the method struggles to identify the correct values for the remaining parameters.

