# OpenReview forum: "Ultra-NeRF: Neural Radiance Fields for Ultrasound Imaging"
_MIDL.io/2023/Conference — MIDL 2023 Oral_

### Official Review · Reviewer_APnR · 2023-01-29

**Confidence:** 4
**Preliminary Rating:** 4
**Recommendation:** Poster

**Summary:**

The authors propose using neural radiance fields (NeRF) to model the interaction of ultrasound waves with tissue that gives rise to BMode images using a neural network, and propose to use this to do volumetric reconstructions of anatomical images by compounding multi-view predicted images. The method is demonstrated in synthetic data and in a phantom where the BMode images and the relative position and orientation of these images is available.

The paper is well written, and proposes a very novel and interesting approach to inverse problem solving in medical ultrasound imaging. The validation is somewhat limited to phantom and synthetic data from one subject each, and although quantitiative data form tissue (attenuation, density etc) can be calculated, this is left out of the validation.

Overall, an interesting proof of concept paper that could benefit from a more thourough validation and a discussion on how this will move forward to real in vivo data.

**Strengths:**

* A novel approach to image reconstruction for ultrasound that combines ultrasound physics, and neural implicit representations. I believe this would be of interest for the community.
* Early results suggest ability to recover physical properties of tissue from sweeps.
* Authors commit to make data available, enabling further research on this which can help move the field forward


**Weaknesses:**

* Figures need to be improved. Figures appear at non intuitive locations. For example, Fig. 3 appears before Fig 1 is mentioned in the text. Also, figure 1 is referred to after figure 2 which should be reversed. Also, caption information is incomplete. For example in Fig. 3, please describe all columns and the relation between the left and right ( I am assuming right is synthetic data but it is not specified). Also there is n discussion on any visual differences in the gt and generated synthetic data. What is the relevance of that synthetic dataset, provided it looks very different to real ultrasound images from a phantom?
* The volumes can be generated by compounding generated images; however, it seems like the learned model should contain the information about the object to be reconstructed within.  Why is compounding necessary, instead of retrieving the object directly from the learned NeRF? Why not use density, which is computed as shown in Fig 5?
* Figure 5, which shows the estimated tissue properties, is described in the discussion instead of in the results. Also, these values are not validated; if gt is not available for the phantom, it should at least be available for the synthetic dataset.
* No discussion on to what extent this method relies on perfect alignment of the input images, and therefore how likely it is that this can be used in real patients and what what type of tracking. Also no discussion on how motion will be dealt with.
* The proposed method reconstruct one subject at a time -is there any space for generalising to multiple subjects/larger datasets, or is this something that will be done subject by subject? How is the amount and diversity of input images affecting the accuracy of the result.

**Deanonymize Review:**

yes

**Paper Type:**

methodological development

**Questions To Address In The Rebuttal:**

I would suggest looking through the list of weaknesses and trying to address those. The main things to address are:
1. Make sure figure organization is consistent: if figures appear near or after they are referenced in the text, this makes reading easier. Also make sure they are in order.
2. Include the physical tissue property estimation in the experimentation and validation. It seems a crucial part of the paper, which is highlighted in the abstract so should be validated to be able to make proper claims

---

### Official Review · Reviewer_f3NT · 2023-02-03

**Confidence:** 3
**Preliminary Rating:** 2

**Summary:**

The author presented a physics-enhanced implicit neural representation (INR) for ultrasound (US) imaging that learns tissue properties from overlapping US sweeps. The presented work is the first to address view-dependent US image synthesis using INR. The author used synthetic and phantom B-mode images datasets to validate the performance.

**Strengths:**

The presented work is the first to address view-dependent US image synthesis using INR. This paper is very innovative. It supports progress towards integrating the implicit 3D US representation exemplified with NeRF into medical applications.

**Weaknesses:**

1. Figure 2 presents an overview of the system of models. Page 5, section 3.2 put some words briefly explains the architecture of the framework.  It is reasonable, due to the page limit, it is difficult to elaborate more in the main context. But, it is worth using more words to clearly present the design of the model, either in the main context or in the appendix.
In addition to this, in Appendix A, It is suggested to provide a complete explanation of your design of the architecture. You cannot only put an addition or complement of Section 3.2.
It is suggested to present, 1) What is the input and output of this architecture? 2) Why did you design this type of model? What are the reasons you selected MLP, not other types of neural network?

2. In Figure 2, $\lambda L_{S S I M}\left(I^{\prime}, I\right)+(1-\lambda) L_2\left(I^{\prime}, I\right)$, is this a loss function?
The reason of designing and choosing this loss function is not clearly demonstrated. The explanation of this loss function is missing in the main context. The explanation of 'Our loss is a weighted sum of SSIM and L2 between the rendered and true B-mode images and controlled by a parameter λ ∈ [0, 1]. In our experiments we used λ = 0.9.' in the appendix is not enough. It is suggested to clearly explain why you choose this loss function and the parameters.

3. In this paper, a 8 layer MLP is selected. Will this cause overfitting? Normally we use 3-4 layers of MLP, and this is enough. 8 layers of MLP is questionable. It is suggested to provide a separate section, called 'hyper-parameter tuning' to show the process you investigate this. Also, Adam optimizer does not need to be cited. It is so commonly used.

4. I wish the author could perform more experiments to show that their methods have good performances. Experimental results and analysis of the results can go into the Appendix section. In the appendix section, It is suggested to explain more on the experiments.



**Deanonymize Review:**

no

**Paper Type:**

methodological development

**Questions To Address In The Rebuttal:**

Please see 'Weakness' section for details. It will be great to explain these in the rebuttal.

In addition to these,

1. In this paper you have presented a work using INR. Why INR is better? Why other methods are not comparable or have potential shortages against INR? It is suggested to clearly demonstrate this and present other methods results as baseline.

2. In this experiment, what is your training and testing dataset? Did you use the same dataset to perform both training and testing?

---

### Official Review · Reviewer_pc1y · 2023-02-05

**Confidence:** 2
**Preliminary Rating:** 3

**Summary:**

The authors propose using NeRF-like architecture for ultrasound imaging. They follow the NeRF settings where the main objective is to obtain renderings from an arbitrary view. In the US context, the NeRF-like model is trained to learn a mapping from 3D to 5D. The authors show that they can synthesize novel views from US sweeps. They show the performance of their method on synthetic data and phantom data.

**Strengths:**

NeRF-like architecture (neural field) is a promissing approach for representing data, and is useful for various modalities. It is therefore interesting to see, how neural representation can be utilized for ultrasound imaging.

**Weaknesses:**

It is unclear what is the main objective of this work: generate novel views or infer 3D structure? The work seems very preliminary. It is only tested on synthetic and phantom data. No comparison with any baseline method is provided.

**Deanonymize Review:**

no

**Paper Type:**

methodological development

**Questions To Address In The Rebuttal:**

How do the results obtained using neural fields compare to some baseline methods? Why focus on rendering novel views? Why not focus on accurately representing some 3D structure? Please also elaborate the role of rendering in your method. It is a bit unclear, as you have NeRF with rendering and NeRF without rendering.

---

### Official Review · Reviewer_yt2F · 2023-02-06

**Confidence:** 4
**Preliminary Rating:** 4
**Recommendation:** Poster

**Summary:**

The paper proposes an Ultrasound (US) physics-inspired rendering module in an implicit neural representation-based method that uses a set of 2D US images for novel view synthesis of target anatomy from any arbitrary viewing direction. For any 3D point in the target anatomy, a neural network first learns to map it to five parameters of a US image formation model. Then a rendering module learns to synthesize a 2D image from these parameters. Tracked 2D US images from a phantom and images synthesized from CT scans are used to train and evaluate the method. Initial results show that the network can synthesize US images, including the properties such as shadows.

**Strengths:**

The idea of estimating parameters related to the US image formation model and the US physics-based rendering module within an end-to-end deep neural network is novel and interesting to my knowledge.

Initial results show that the method is able to synthesize various aspects of an ultrasound image, such as shadows.

**Weaknesses:**

- Some aspects of the paper are difficult to understand, such as the figures and the details on how the network is trained (see detailed comments).

- The experiments are carried out on phantom and synthetic data that do not consider the real-world scenario of motion of target anatomy. It is unclear how exactly the novel view synthesis would be used in real-life scenarios, and whether the method can be trained with moving target anatomy.

- The main contribution of the paper is to learn the US physics-inspired parameters, but there are very limited results (single figure with little explanation) showing whether these parameters estimated by the network consistently correspond to the expected properties across several images.

**Deanonymize Review:**

no

**Detailed Comments:**

Anisotropy in ultrasound: the term is not well-defined in the paper; it would be helpful for the readers to define it in the paper’s context.

Fig 1 (b) is difficult to understand. The captions (or labels in the image) should describe what the three rows and columns represent. The first row seemed to be compounded images, but then it seems to be one 2D view, as the term compounding in this paper seems to have been used only for the 3D surface view. I couldn’t understand the relationship between the angles and colors. The mid-row seems to show TSDF representation of the anatomy. Does it represent only the surface of a target anatomy or the whole volume? I.e. what would slices of the TSDF look like? From the figure, it seems it only represents the surface of the white structure seen in the first row.

Fig 5. What values do the colors correspond to? The figure needs a color bar.

There is limited analysis of Fig 5 in the discussion section. Since rendering with US physics parameters is the main contribution of the paper, it would be better to provide a more detailed analysis (with a few more randomly sampled images) on whether the estimated parameters are consistent to what they are meant to represent. For example, beta seems to be high almost everywhere (if yellow/whitish > blue/blackish). The dark regions in the US image can represent shadows or fluid (it seems there are both in Fig 5 as well; a vertical shadow and an elliptical fluid region), but both seem to have similar values for all the parameters except for scattering density and amplitude.

What are t_n & t_f in Eq 1&2? What’s “d” in Eq 3, perhaps “t”?

Is the rendering module fully differentiable for the loss to backpropagate?

**Paper Type:**

methodological development

**Questions To Address In The Rebuttal:**

The paper’s presentation can be improved, especially by labeling the figures and improving the captions. I’d like to know how this method would translate to real-word scenarios of moving target anatomy. A rigorous qualitative analysis of the consistency of the parameters estimated by the network for a set of randomly sampled images synthesized by the network would help better assess the method's relationship with the US image formation model.

*Final rating after rebuttal*
The core idea of the paper is in introducing a set of parameters intended to be mapped to known physical parameters. There were limited experiments dedicated to understand whether these parameters are being learned as intended. The authors have added some new observations in appendix. It seems that the parameters do not necessarily learn as intended and it needs further study. Nevertheless, addition of the flexible but limited parameters in the NERF setting could enable future work along this line where parameters aligned with physical properties are learned.
I've raised the scoring from the original rating.

---

### Meta-Review · Area_Chair_sG4g · 2023-02-24

**Recommendation:** Accept (Poster)
**Confidence:** 4

**Metareview:**

The authors propose using NeRF-like architecture to obtain renderings from an arbitrary view of ultrasound imaging. The method is validated on synthetic data and on a phantom. While results on real data would have been a plus, this first feasibility study is interesting.